# Parameter Optimization Model Photovoltaic Battery System for Charging Electric Cars

**Peter Tauš [1],*** , **Marcela Taušová [1]** , **Peter Sivák [1]** , **Mária Shejbalová Muchová [1]** and **Eva Mihaliková [2]**

[1] Institute of Earth Sources, Faculty of Mining, Ecology, Process Technologies and Geotechnology, Technical University of Košice, Letná 9, 042 00 Košice, Slovakia; marcela.tausova@tuke.sk (M.T.); peter.sivak.2@tuke.sk (P.S.); maria.shejbalova.muchova@tuke.sk (M.S.M.)

[2] Department of Economics and Management of Public Administration, Faculty of Public Administration, Pavol Jozef Šafárik University in Košice, Popradská 66, 041 32 Košice, Slovakia; eva.mihalikova1@upjs.sk

* Correspondence: peter.taus@tuke.sk

**Abstract:** Sales of electric cars and vehicles (EVs) have recently been showing a rapidly growing trend. In connection with rising electricity prices as well as social pressure on the environmental impacts of electromobility, there is also increasing interest of EV owners in the ecological source of electricity. The largest group of owners of EVs are residents of family houses, so, logically, they focus their attention on the possibility of using photovoltaic (PV) charging systems for EV charging. The design of the PV system for supporting EV charging is problematic due to several input parameters in the calculation of energy needs and due to the inconsistencies of electricity generation with normal electric vehicle (EV) charging time. While the PV system produces electricity during the day, family homeowners require charging EVs mainly at night. This requires batteries as part of a PV system. The optimal design of the PV of the battery system must take into account the real consumption of EV, the average daily distance traveled, the location, the weather bridging time, and, last but not least, the investor's financial situation. The timing mismatch of electricity needs and generation may result in the oversizing or sub-scaling of the PV system depending on the time period for which the investor claims full coverage. With an average daily EV consumption of 10 kWh/day, the overproduction of electricity may be at 8620 kWh per year if it is required to fully cover PV systems in January. Conversely, for the installation of PVs for full coverage in August, the year-round electricity deficit will be 1500 kWh per year. For the analyzed geographical conditions, i.e., Latitude 48.8, the optimum performance of PV system for one-day electricity storage is 3.585 kW. This corresponds to the full coverage of EV consumption in March, the price of the whole system varies from EUR 9000 to EUR 20,000 depending on the type of battery. In addition to the battery price, the required accumulation time for electricity to overcome adverse weather increases the required performance of a photovoltaic system (PVS), which again results in system overshooting and financial loss by not using the generated electricity. This cycle of interdependencies is usually very difficult to adjust optimally. In the contribution, we analyzed the mutual relationships of calculating the performance of a PVS according to the daily consumption of EV and required time of overcoming adverse weather. The input data for the analyses were normal average EV consumption and the number of daily km traveled from 10 to 100 km/day scaled to 10. The optimization process consisted of determining the necessary performance of the PVS and its production in the event of a requirement to ensure full energy demand in each month. In addition, different types of batteries that influence the investment price enter into optimization analyses. This depends on the energy density of a given battery, the depth of discharge, capacity, and type. The result of this research is a computational model for determining a new indicator—we called it the monthly deviation factor. This indicates the degree of oversizing or undersizing of the PV system in relation to the stated factors.



**Keywords:** photovoltaic systems; electric cars charging; battery system; economy; renewable energy sources; energy efficiency

## 1. Introduction

The current state of energy and its environmental impact is one of the most discussed topics around the world. There are various scenarios and targets for reducing greenhouse gas emissions, tail pipe emissions that are specific to vehicles with internal combustion engines, improving the environment and increasing the efficiency of energy production and use. Most scenarios share three steps—(1) reducing energy demand, thus reducing consumption, (2) the use of Renewable energy sources (RES) wherever technically possible and (3) complementing the remaining energy needs with clean technologies [1]. The main problem is the optimal alignment of two conflicting energy problems—despite national objectives and efforts, there is an increasing demand for energy and efforts to reduce the negative impact of energy on the environment [2]. Currently, the most important energy source that meets both conditions are renewable energy source. The advantage is that, with RES, we can produce heat and electricity. The main disadvantages of most RES are their unpredictability and the instability of their production. Many governments are trying to increase the share of RES in the overall energy mix by various forms of grants and support. Scientific analyses and simulations indicate that an optimal system of subsidies to RES discourages consumers and energy producers from fossil fuels, which has a positive impact on emissions and balance of energy in the state [3]. An integral part of the energy mix will be electric and thermal energy accumulators due to the time volatility of RES. Thermal energy does not require immediate deliveries of huge outputs in short periods of time in the form of peak energy. On the contrary, electricity is characterized by increased demand almost immediately based on the load curve of the electrification system. This needs to be provided either through a robust energy grid or suitably dimensioned electricity batteries in combination with RES. Such an energy system is called hybrid. It is a combination of different energy carriers and energy storage systems into a single (state) whole. The combination of different technologies, such as heat and electricity generation through cogeneration, heat pumps, photovoltaic systems, etc., facilitate the integration of different energy carriers into a single energy unit, thereby increasing its flexibility and overall efficiency [4,5].

Apart from industry, one of the main air pollutants is emitted by internal combustion engine vehicles. The solution is to increase the share of electric cars in transport, on the condition that they use as much electricity produced from RES as possible. Over the past few years, it has been proved that, with the use of electric vehicles (EVs), individual states, but mainly drivers, are serious. In 2017, more than one million EVs were sold worldwide, an increase of 50% compared to 2016. The increase in the number of EVs on the market is being tackled in a variety of ways in different countries. China has the largest share of EV sales, where more than 580,000 EVs were sold in 2017. Norway, in turn, is the country with the largest share of EVs, with up to 39% of electric cars sold there in 2017 [6]. As can be seen, EV's share in the car market is constantly increasing. When considering buying an EV, one of the main assessment factors is the EV range and the energy consumption in kWh/100 km. Research on the assessment of the real consumption of EVs focuses on different areas. Interesting results came from Ahn K. research [7]. This states that EV consumption depends so strongly on driving patterns that it is necessary to consider changing the coordination of traffic signs. By optimizing it, EV energy efficiency can be improved. It is clear that total annual energy consumption is an important feature of EVs. It is also related to the operating costs of EVs. EV owners are therefore looking for ways to reduce operating costs, i.e., charging the EV. On the other hand, energy network operators are looking for ways to optimize the electricity consumption profile. The problems with the energy network associated with the immediate increase in electricity consumption are becoming more pronounced in connection with the increase in the number of EVs and the need to charge them.

One of the technically and economically available solutions is the use of photovoltaic (PV) system with accumulation as the primary source of EV charging. However, such a system needs to be designed and operated very sophisticatedly. Research into the elimination of negative impacts of charging stations on the distribution network is very intensive. Hussain S. in his study [4] proposes a Fuzzy Logic (FLWCS) charging scheme for EV charging. This optimally divides the charging power between charged EVs in an optimized way. Research [8] shows that an effective method for optimizing consumption and charging for electric cars can be a hybrid optimization algorithm through which the system learns and optimizes charging for a given location and its specific conditions. The algorithm considers three basic variables that change in real-time: real-time electricity price in different price bands, real-time calculation of PV energy from solar radiation data, and optimization to minimize the operating costs of the EV charging station.

Intelligent charging system for parking with integrated PV system aims [9] to create a common model for planning the energy flow between the grid, energy storage system (ESS) and PV charging system. On this basis, it was found that, for EVs that regularly drive at a certain interval, it is possible to create a rather precise prediction and better prepare a charging plan for the next 24 h.

Rücker F. [10] focused his research on smart charging strategies in conjunction with PV systems. It found that, with the support of PV systems, the consumption of electricity from the EV charging network in households decreased by up to 48.1% and electricity costs decreased by 17.6% per year. Research also focuses on the impact of smart charging in combination with a PV battery life system. The optimization of EV charging from the PV system is also researched [11,12]. It notes that multi-objective optimization is an appropriate method. The aim is to minimize the consumption of solar energy and to charge the electric vehicle appropriately. The linear programming method can also be used for optimization.

Clearly, in addition to intelligent traffic control of the PV charging system EV, great attention must be paid to its dimensioning in such a way that not only the investment but also the operation in combination with the distribution network are optimized for the consumer.

The aim of the research was to create a model for the implementation of more extensive calculations and analyzes in the conditions of various European countries and to verify its functionality. The operating parameters of solar systems can be estimated with a certain tolerance, taking into account the instability of solar conditions. The dynamism of the model lies in its use by input specific parameters for the specific conditions of a given country at a given time.

## 2. Materials and Methods

### 2.1. Entry Data and Design of the Computational Model

In terms of input data for the calculation of the parameters of the PV system, the first step was the determination of solar radiation parameters. In determining the required performance of the PVS and the installation area, we have considered the values of the radiation for Latitude 48.8 in Europe, which is the mean annual value for European countries of 1240 kWh·m$^{-2}$. This figure is the average for the years 2010–2016 obtained from the PVGIS web application (Photovoltaic Geographical Information System, Version: 5.1, European Commission Joint Research Centre, at the JRC site in Ispra, Italy) [13].

Energy consumption of EVs depends on several factors, in research, we have considered the following:

- The average distance traveled by EV per day;
- Mean EV consumption per day;
- The way of driving;
- The type of user;
- The user's location;
- Engine power;
- Vehicle load.

The basis for establishing EV energy consumption intervals (EC$_{EV}$) was the results of the tests according to the methodology established by EPA (Environmental Protection Agency). The minimum consumption was 14.9 kWh/100 km–Tesla Model 3 Standard Range Plus 2020. The maximum consumption was 31.1 kWh/100 km–Porsche Taycan Turbo S 2020 [14]. For analysis, therefore, intervals with a range of 10 kWh/100 km were created in the range of minimum and maximum, rounded to tens below. All symbols are explained on first use or in Table 1.

The calculation of the total average consumption of EV per day $EC_D$ was calculated using:

$$EC_d = \frac{1}{100} \cdot EC_{EV} \cdot D_d \left( \frac{kWh}{day} \right) \tag{1}$$

$D_d$—the number of kilometers traveled by EV per day has also been determined at intervals the minimum and maximum of which has been determined by research in the area. Long-term research suggests that 75% of the car users drive equal to or less than 40 km [15].

The individual driving distance distribution of weekdays are shown in Figure 1. The given data are valid for Denmark, because it is one of the developed European countries with a high share of RES in the energy mix and this is the EU's goal for all European country.

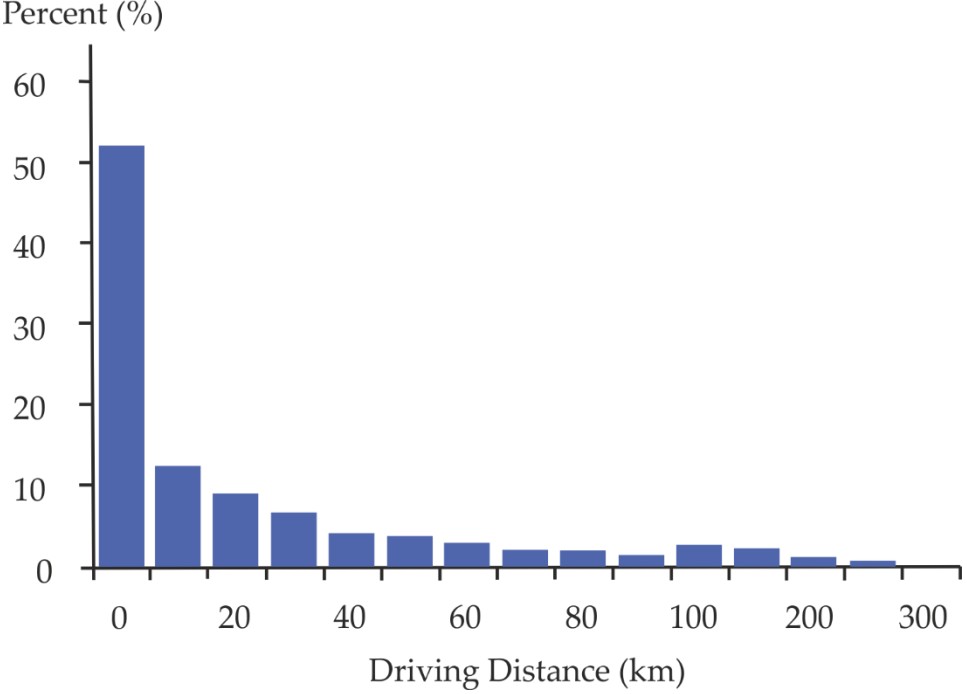

**Figure 1.** Individual average driving distance distribution of weekdays [15].

The number of days of accumulation is often considered in the calculations of the performance of the PV system. This represents the maximum time to bridge the lack of solar energy. In case of accumulation of electricity into batteries for charging in time without sunshine, the size of the accumulation system should be considered. As its capacity increases, the necessary performance of the PV system increases, while the price increases. As part of our research, we considered the case that the electricity produced by the PV system is either delivered directly to the EV battery or the storage system. That means we are not thinking about overcoming bad weather. The battery size is set for the storage of electricity produced by the PV system for one day. We are considering a hybrid system connected to the network, so increasing battery capacity results in a deterioration of the system's economic parameters. In the analyzes and modeling, we considered batteries suitable and commonly used in PV battery systems. The considered PV charging system will consist of a PV generator, a charge

controller, a battery. The electricity obtained from the PV system is either stored directly in the battery of the electric vehicle (in case it is present during the day) or is stored in the solar battery of the PV system. From it, the electric car is subsequently charged at night or at times without sunlight. Input data for data calculation for statistical processing are shown in Table 1.

**Table 1.** Input data for data calculation for statistical processing.

| | | |
|---|---:|---|
| $\tau$—Number of days for accumulation | 3 | - |
| $\eta_{PVP}$—Efficiency of PV pannel | 0.18 | (%) |
| $\eta_{PVS}$—Efficiency of PV system | 0.86 | (%) |
| $\eta_{PVT}$—Total Efficiency | 0.1548 | (%) |
| $D_d$—Distance per Day | 10 | (km/day) |
| $EC_d$—Total consumption per Day | | (kWh/day) |
| $A_{PV}$—Area of PV field | | (m$^2$) |
| $H_{Md}$—Daily average irradiation in given Month | 1.045 | (kWh/m$^2$.day) |
| $P_{UPV}$—Output Unit Power of PV system | 154.8 | W/m$^2$ |
| $P_{PV}$—Output Power of PV system | | kWp |
| $U$—Voltage of system | 12 | V |
| $T_{DOD}$—Depth of battery discharge | 0.8 | (%) |
| $\eta_{INV}$—Efficiency of Inverter | 0.95 | (%) |
| $\eta_D$—Efficiency of energy distribution | 0.98 | (%) |
| $Q_{Bat}$—Capacity of battery | | Ah |
| $C$—Cost of PV system | | € |
| $C_{BAT}$—Battery cost | 1.5564 | €/Ah |
| Battery Type | LTO | |

Legend:

| | |
|---|---|
| | Cells with selectable range of values |
| | Average PV values valid for polycrystalline PV panels (Can be changed) |
| | Values entered for a specific technology |
| | Calculated values |

The area of the PV field ($A_{PV}$) was calculated by reference to:

$$A_{PV} = \frac{EC_d}{\eta_{PVT} \cdot H_{Md}} \left(m^2\right) \tag{2}$$

$\eta_{PVT}$—total efficiency of PV technology.
$H_{Md}$—daily average irradiation in a given month.

As part of the calculations, we considered a value of 15.48%, which corresponds to the efficiency of the PV panels of 18% and the PV system losses of 14%.

The efficiency of the PV panels and components of the PV system also depend on its unit performance in relation to the surface area of PV panels and STC (Standard Test Conditions). It is calculated according to the relationship:

$$P_{UPV} = 1000 \cdot \eta_{PVT} \left(Wp/m^2\right) \tag{3}$$

The total required installed power PV of the system for the selected daily electricity consumption is calculated according to the relationship:

$$P_{PV} = P_{UPV} \cdot A_{PV} (Wp) \tag{4}$$

In the absence of EV at the time of the PV system's electricity generation system, the energy must be stored in the battery. A variety of factors influence the overall required battery capacity. The most important are:

$U$—voltage of the system (V);

$T_{DOD}$—depth of battery discharge (%);

$\eta_{INV}$—efficiency of the inverter (%);

$\eta_D$—efficiency of energy distribution (%).

The required battery capacity is calculated according to the relationship [16]:

$$Q_{BAT} = \frac{1000 \cdot EC_d \cdot \tau}{U \cdot T_{DOD} \cdot \eta_{INV} \cdot \eta_D} (Ah) \qquad (5)$$

Other battery parameters are also important for the EV owner, such as

$ED_{BAT}$—energy density of battery (Wh/kg);

$M_{BAT}$—battery weight (kg);

$C_{BAT}$—battery cost (EUR/Ah).

The parameters entering the calculation model also take into account the required operating parameters, such as battery life and battery efficiency. They are listed in next text and tables. Within the research [5], these parameters are considered in the variable $\eta_D$-Efficiency of energy distribution (Table 1).

Since the objective of the research was to optimize the PV system with accumulation, it was necessary to create a computational model for obtaining data for statistical processing, taking into account all variables in each step. For the above parameters and their context, we created a calculation model in MS Excel with the input data given in the table.

Calculations have been made for all variants of the combinations of specified input data ranges—Table 1. Variants are calculated by selecting the range from the green cells listed in Table 2. $H_{Md}$ values were calculated for each month based on data from the PVGIS program for the selected geographical area of Europe [13].

**Table 2.** Ranges of variables used in individual calculation steps.

| $D_d$ (km/day) | Month | $H_{Mm}$ (kWh/month) | $H_{Md}$ (kWh/day) |
|---|---|---|---|
| 10 | I. | 32.38 | 1.04 |
| 20 | II. | 51.07 | 1.82 |
| 30 | III. | 56.41 | 1.82 |
| 40 | IV. | 81.72 | 2.72 |
| 50 | V. | 129.28 | 4.17 |
| 60 | VI. | 163.92 | 5.46 |
| 70 | VII. | 167.99 | 5.42 |
| 80 | VIII. | 155.44 | 5.01 |
| 90 | IX. | 157.56 | 5.25 |
| 100 | X. | 143.54 | 4.63 |
| | XI. | 61.16 | 2.04 |
| | XII. | 50.63 | 1.63 |

*2.2. Calculation of Parameters for Statistical Data Analysis*

The results of each calculation step were the values of quantities entering the next phase—the analysis of numerical variables using descriptive statistics are shown in Table 3. The total number of data for these input data was 3120. It is the sum number of the number of outputs from the created model for the combination of all inputs for one type of PV system (standard configuration in Europe) and for the two mentioned battery types. For one input listed in Table 1, the output of the model is 26 outputs (Table 2). Statistically, it is necessary to process for a given data system configuration 3120 for one PV charging system with one battery type.

**Table 3.** Results of one step of calculations for selected input conditions.

| $EC_{EV}$ | $D_d$ | $EC_d$ | $A_{PV}$ | $H_{Md}$ | $P_{PV}$ | $Q_{bat}$ | | $M_{Bat}$ | $C$ |
|---|---|---|---|---|---|---|---|---|---|
| (kWh/100 km) | (km/day) | (kWh/day) | (m²) | (kWh/day) | kWp | Ah | kWh | kg | € |
| 5 | 10 | 0.50 | 3.09 | 1.045 | 0.56 | 55.9 | 0.7 | 9.1 | 1089 |
| 6 | 10 | 0.60 | 3.71 | 1.045 | 0.67 | 67.1 | 0.8 | 10.9 | 1307 |
| 7 | 10 | 0.70 | 4.33 | 1.045 | 0.78 | 78.3 | 0.9 | 12.8 | 1525 |
| 8 | 10 | 0.80 | 4.95 | 1.045 | 0.89 | 89.5 | 1.1 | 14.6 | 1742 |
| 9 | 10 | 0.90 | 5.57 | 1.045 | 1.00 | 100.7 | 1.2 | 16.4 | 1960 |
| 10 | 10 | 1.00 | 6.18 | 1.045 | 1.11 | 111.9 | 1.3 | 18.2 | 2178 |
| 11 | 10 | 1.10 | 6.80 | 1.045 | 1.22 | 123.1 | 1.5 | 20.1 | 2396 |
| 12 | 10 | 1.20 | 7.42 | 1.045 | 1.34 | 134.3 | 1.6 | 21.9 | 2614 |
| 13 | 10 | 1.30 | 8.04 | 1.045 | 1.45 | 145.5 | 1.7 | 23.7 | 2831 |
| 14 | 10 | 1.40 | 8.66 | 1.045 | 1.56 | 156.6 | 1.9 | 25.5 | 3049 |
| 15 | 10 | 1.50 | 9.28 | 1.045 | 1.67 | 167.8 | 2.0 | 27.4 | 3267 |
| 16 | 10 | 1.60 | 9.90 | 1.045 | 1.78 | 179.0 | 2.1 | 29.2 | 3485 |
| 17 | 10 | 1.70 | 10.51 | 1.045 | 1.89 | 190.2 | 2.3 | 31.0 | 3703 |
| 18 | 10 | 1.80 | 11.13 | 1.045 | 2.00 | 201.4 | 2.4 | 32.8 | 3920 |
| 19 | 10 | 1.90 | 11.75 | 1.045 | 2.12 | 212.6 | 2.6 | 34.7 | 4138 |
| 20 | 10 | 2.00 | 12.37 | 1.045 | 2.23 | 223.8 | 2.7 | 36.5 | 4356 |
| 21 | 10 | 2.10 | 12.99 | 1.045 | 2.34 | 235.0 | 2.8 | 38.3 | 4574 |
| 22 | 10 | 2.20 | 13.61 | 1.045 | 2.45 | 246.2 | 3.0 | 40.1 | 4792 |
| 23 | 10 | 2.30 | 14.22 | 1.045 | 2.56 | 257.3 | 3.1 | 42.0 | 5009 |
| 24 | 10 | 2.40 | 14.84 | 1.045 | 2.67 | 268.5 | 3.2 | 43.8 | 5227 |
| 25 | 10 | 2.50 | 15.46 | 1.045 | 2.78 | 279.7 | 3.4 | 45.6 | 5445 |
| 26 | 10 | 2.60 | 16.08 | 1.045 | 2.89 | 290.9 | 3.5 | 47.4 | 5663 |
| 27 | 10 | 2.70 | 16.70 | 1.045 | 3.01 | 302.1 | 3.6 | 49.3 | 5880 |
| 28 | 10 | 2.80 | 17.32 | 1.045 | 3.12 | 313.3 | 3.8 | 51.1 | 6098 |
| 29 | 10 | 2.90 | 17.94 | 1.045 | 3.23 | 324.5 | 3.9 | 52.9 | 6316 |
| 30 | 10 | 3.00 | 18.55 | 1.045 | 3.34 | 335.7 | 4.0 | 54.7 | 6534 |

The aim was to find and define the dependencies of individual variables entering the process of selecting a PV system with accumulation for charging EV. The method developed, thus allows instant calculation of parameters for a specific PV technology, battery and EV. The values of these parameters are inserted into the model into yellow cells. An example is the calculation result for polycrystalline PV technology, the LTO—Lithium Titanate Oxid Battery Cell battery (the parameters are highlighted in Table 5 and the input conditions are according to Table 1).

For illustration, we list the selected results of the calculations of the proposed model for the changed values in Table 4:

$D_d$ = 40 km/day

$H_{MD}$ = 5.464 kWh/day (June)

**Table 4.** Results of one step of calculations for changed input conditions.

| $EC_{EV}$ | $D_d$ | $EC_d$ | $A_{PV}$ | $H_{Md}$ | $P_{PV}$ | $Q_{bat}$ | | $M_{Bat}$ | $C$ |
|---|---|---|---|---|---|---|---|---|---|
| (kWh/100 km) | (km/day) | (kWh/day) | (m²) | (kWh/day) | kWp | Ah | kWh | kg | € |
| 5 | 40 | 2.00 | 2.36 | 5.464 | 0.43 | 223.8 | 2.7 | 36.5 | 1114 |
| 10 | 40 | 4.00 | 4.73 | 5.464 | 0.85 | 447.5 | 5.4 | 73.0 | 2229 |
| 20 | 40 | 8.00 | 9.46 | 5.464 | 1.70 | 895.1 | 10.7 | 145.9 | 4458 |
| 30 | 40 | 12.00 | 14.19 | 5.464 | 2.55 | 1342.6 | 16.1 | 218.9 | 6686 |

Battery parameters derived from the results of research in the field were used in the calculation. The main factors influencing the calculation were price, capacity, charging costs and weight of batteries. Selected parameters used in the calculations are listed in Table 5 [16].

**Table 5.** Battery parameters used in calculations. Data from [16].

| Parameter | Unit | Battery Type | | | | | | |
| | | Pb | Pb | Pb | Pb | Pb | LiNiMnCoO2 | Li4Ti5O12 |
| | | Traction | Traction | Traction | Station 4–6 Years | Starter Batteries | (NMC) | (LTO) |
| Capacity | (Ah) | 60 | 115 | 150 | 75 | 56 | 94 | 40 |
| | (Wh) | 720 | 1380 | 1800 | 900 | 672 | 338 | 92 |
| Voltage | (V) | 12 | 12 | 12 | 12 | 12 | 3.6 | 2.3 |
| Number of cycles at 50% DOD | (-) | 200 | 800 | 800 | 250 | 500 | 1000 | 20,000 |
| Price | (€) | 81.4 | 272.2 | 304.9 | 158.5 | 70.1 | 148.2 | 62.3 |
| $\eta_{SH}$ | (%) | 60 | 60 | 60 | 60 | 60 | 85 | 85 |
| $\eta_{HH}$ | (%) | 91 | 91 | 91 | 91 | 91 | 95 | 95 |
| $ED_{BAT}$ | (Wh/kg) | 720 | 1380 | 1800 | 900 | 0.4–1.2 | 150–220 | 50–80 |
| * Price for 1 kWh at $\eta_{SH}$ | (€/kWh) | 0.942 | 0.411 | 0.353 | 1.174 | 0.348 | 0.516 | 0.040 |
| * Price for 1 kWH at $\eta_{HH}$ | (€/kWh) | 0.621 | 0.271 | 0.233 | 0.774 | 0.229 | 0.461 | 0.036 |
| * Average Price ($AP_{BAT}$) for 1 kWh | (€/kWh) | 0.781 | 0.341 | 0.293 | 0.974 | 0.288 | 0.488 | 0.038 |

*\* Price of electricity from a battery charged by a PV system*

The resulting economic parameters of the PV system depend primarily on the ability of efficient consumption of electricity produced by the PV system and from system management operational dispatching. According to [17], it is possible to increase energy savings from the PV system by up to 45% by suitable technical and economic dimensioning; the demand for PV energy has decreased by up to 10% and the need for a backup source has been reduced by 92%. Even research has shown that the annual capacity of the battery will increase by up to 10%. In our research, we focused on the PV system used exclusively for charging EV with battery support. Different variants of calculations have one common problem. This is the determination of the optimal performance of the PV system and the size of the battery in terms of the economics of operation. We analyzed the system from the point of view of the investment costs of the generator, the battery and from the point of view of the EV charging costs. We looked for the optimal cost ratio for electricity purchases from the grid and electricity generation and storage of the proposed PV system.

It is known that the PV system designed to produce 100% of the energy demand in the summer months will not have sufficient performance in winter. This will result in a higher share of electricity from the network and an increase in operating costs. Conversely, the PV system designed to cover 100% of electricity needs in the winter months will have a significant overproduction of electricity in the summer. This will translate into high investment costs, especially in case of the impossibility of supplying excess electricity to the grid, as is the case in Slovakia, for example. An appropriate design of the PV system performance and battery size according to the user's requirements can optimize investment and operating costs in combination with the use of electricity from the grid.

In this part of the research, we determined the annual production of PV charging system with accumulation $EP_Y$ (kWh/year) according to the specified input parameters EV, PV, and accumulator. The result was the rate of oversized or subdivision of the PV system in individual months. We called this calculated measure a "monthly deviation factor $f_{DM}$" and it was calculated by:

$$f_{DM} = \frac{EP_Y}{EC_Y} \; (-) \tag{6}$$

$EP_Y$—annual production of PV charging system with accumulation (kWh/year).

$EC_Y$—annual electric car consumption (kWh/year).

Approaching factor to 1 means optimizing the design of the PV charging system with accumulation for specific conditions and specifically EV. Based on this indicator, it was possible to define an economically optimal PV charging system EV. This occurs in the case $f_{DM}$ = 1. For this purpose, statistical analyses of the data obtained had to be carried out.

## 2.3. Analysis of Data

The analysis of numerical variables using descriptive statistics established the basic statistical characteristics for each indicator—mean, Std Dev, minimum, maximum, and coefficient of variability (CV) are shown in Table 6.

**Table 6.** Basic statistical characteristics of numerical variables.

| Parameter | N | Mean | Std Dev | Sum | Min | Max | CV |
|---|---|---|---|---|---|---|---|
| $EC_{EV}$ (kWh/100 km) | 3120 | 17.50 | 7.50 | 54,600.00 | 5.00 | 30.00 | 42.86 |
| $D_d$ (km/day) | 3120 | 55.00 | 28.73 | 171,600.00 | 10.00 | 100.00 | 52.23 |
| $H_{Md}$ (kWh/day) | 3120 | 3.42 | 1.64 | 10,668.81 | 1.04 | 5.46 | 48.09 |
| $EC_d$ (kWh/day) | 3120 | 9.63 | 6.85 | 30,030.00 | 0.50 | 30.00 | 71.18 |
| $A_{PV}$ (m$^2$) | 3120 | 24.55 | 24.88 | 76,598.66 | 0.59 | 185.54 | 101.35 |
| $P_{PV}$ (kWp) | 3120 | 4.42 | 4.48 | 13,787.76 | 0.11 | 33.40 | 101.35 |
| $Q_{BAT}$ (Ah) | 3120 | 1076.91 | 766.54 | 3359,962.41 | 55.94 | 3356.61 | 71.18 |
| $Q_{BAT}$ (kWh) | 3120 | 12.92 | 9.20 | 40,319.55 | 0.67 | 40.28 | 71.18 |
| $M_{BAT}$ (kg) | 3120 | 175.58 | 124.98 | 547,819.96 | 9.12 | 547.27 | 71.18 |
| $C$ (€) | 3120 | 9630.58 | 8940.21 | 30,047,412.10 | 278.60 | 65,338.85 | 92.83 |
| $EC_Y$ (kWh/year) | 3120 | 3513.13 | 2500.63 | 10,960,950.00 | 182.50 | 10,950.00 | 71.18 |

Coefficient of variation (CV) values indicate high variability of variables–$EC_D$, $A_{PV}$, $P_{PV}$, $Q_{BAT}$, $M_{BAT}$ and $C$, some of which have a volatility of up to 100%. We further explored what causes such a high variability.

**ANOVA: Analysis of variability by month**

Using the ANOVA analysis, we assessed the variability of the values of selected variables by the impact of months. This was confirmed to be statistically significant in the three variables—the required area ($A_{PV}$), the required performance of the $P_{PV}$, and the price of the PV system, which is directly related to the area (Figure 2). For these three indicators, the Prob > F value is less than 0.0001, which is lower than the significance level alpha = 0.05, which means that the difference between the average values in the individual months is statistically significant. The partial indicators of the analysis (degree of freedom (DF), Sum of Squares, Mean Square, F Ratio, Prob > F) are explained below Figure 2.

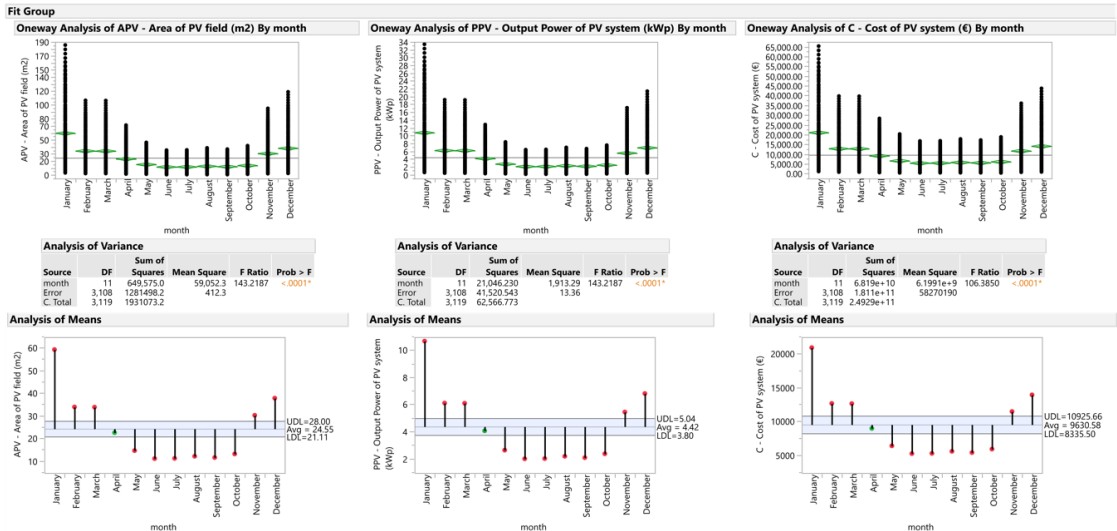

**Figure 2.** Results of analysis of variability due to months.

DF records an associated degree of freedom (DF for short) for each source of variation. The degrees of freedom for C. The total is N-1, where N is the total number of observations used in the analysis. If the X factor has k levels, then the model has k-1 degrees of freedom. The error degrees of freedom is the difference between the C. The total degrees of freedom and the model degrees of freedom (in other words, N-k) [18].

Sum of Squares-Records a sum of squares (SS for short) for each source of variation. The total (C. Total) sum of squares of each response from the overall response mean. The C. Total sum of squares is the base model used for comparison with all other models. The sum of squared distances from each point to its respective group mean. This is the remaining unexplained Error (residual) SS after fitting the analysis of variance model. The total SS minus the error SS gives the sum of squares attributed to the model. This tells you how much of the total variation is explained by the model.

The mean square is the sum of squares divided by its associated degrees of freedom. The model mean square estimates the variance of the error, but only under the hypothesis that the group means are equal. The error mean square estimates the variance of the error term independently of the model mean square and is unconditioned by any model hypothesis [18].

The F Ratio is the model mean square divided by the error mean square. If the hypothesis that the group means are equal (there is no real difference between them) is true, then both the mean square for error and the mean square for model estimate the error variance. Their ratio has an F distribution. If the analysis of variance model results in a significant reduction in variation from the total, the F ratio is higher than expected [18].

Prob > F-Probability of obtaining (by chance alone) an F value greater than the one calculated if, in reality, there is no difference in the population group means. Observed significance probabilities of 0.05 or less are often considered evidence that there are differences in the group means [18].

The results show significantly higher values of indicators in the winter months of November–January and significantly lower values in the summer months of May–September. In the month of April, all three indicators are valued at the average of the year. This month was established as the reference for determining the economic optimism of the installed power of the PV system. We were based on annual electricity requirements defined according to EV consumption.

### 2.4. Creation of a New Indicator—A Monthly Deviation Factor

The annual production $EC_Y$ indicates how much kWh electricity is produced over a **whole year from** a PV system that has been designed to cover 100% of electricity needs in a particular **month, taking into** account climatic conditions. On this basis, we distinguish between yearly production for

design conditions in January-December, with the highest levels reached in January and December, as the system is significantly oversized. If the system was designed according to the need for the summer months' production, e.g., August, the system would be strongly undersized and produce the lowest values in Figure 3.

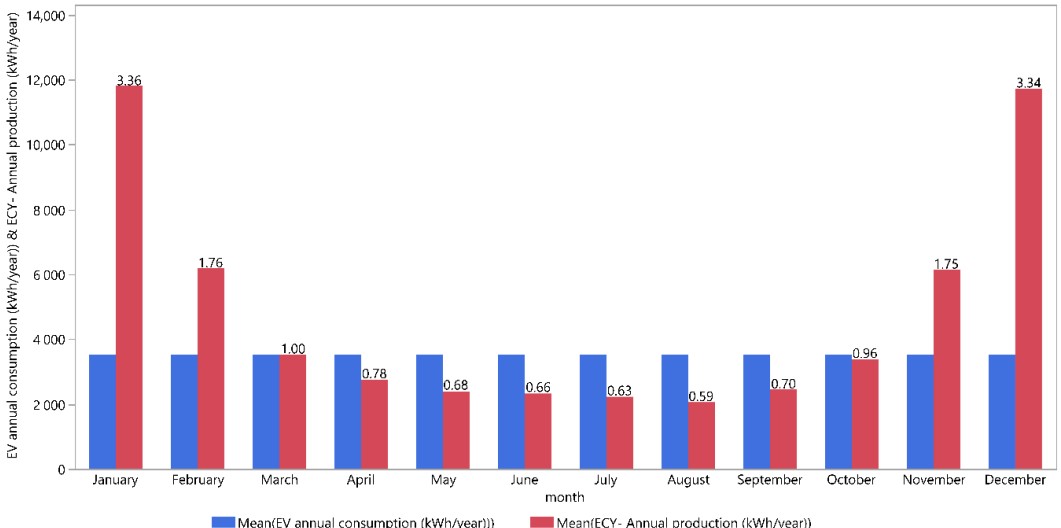

**Figure 3.** Monthly deviation factor versus annual energy demand from January to December.

The results of the analysis show that the smallest variations in oversizing and sub-dimension of the PV charging system are achievable year-round with an average daily consumption of EV not exceeding 3.40 kWh/day (Figure 4). Consumption up to 6.30 kWh/day is economically acceptable. With the growth of daily consumption, the size of the PV system is more demanding, as $f_{DM}$ values increase, which has a negative impact on the economy of such a system.

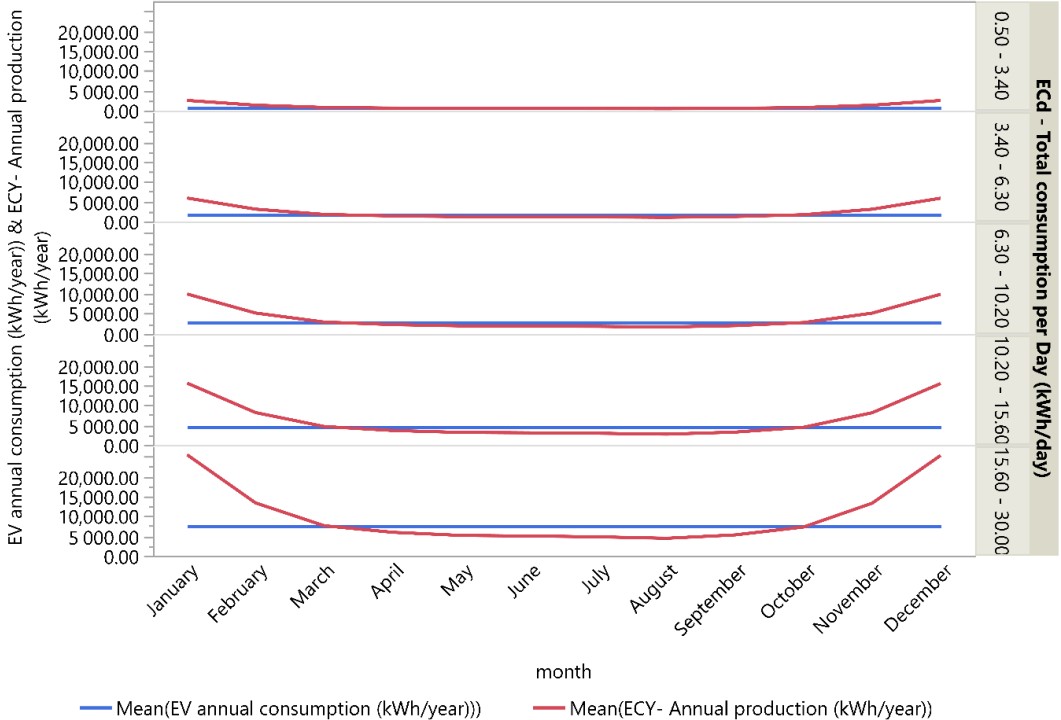

**Figure 4.** Monthly deviation factor versus annual energy demand from January to December according to daily consumption.

## 3. Results and Discussion

In the case of daily distance traveled by EV within 10 km/day, all EV analyzed (Table 3) meet the economic efficiency condition of the PV system. If the EV owner travels 20 km a day, for optimal use of the PV charging system, the consumption of EV should not exceed 17 kWh/100 km (Table 7).

**Table 7.** EV consumption limit value for optimal use of the PV system at a daily distance of up to 20 km/day.

| $EC_{EV}$ | $D_d$ | $EC_d$ | $A_{PV}$ | $H_{Md}$ | $P_{PV}$ | $Q_{bat}$ | | $M_{Bat}$ | $C$ |
|---|---|---|---|---|---|---|---|---|---|
| (kWh/100 km) | (km/day) | (kWh/day) | (m²) | (kWh/day) | kWp | Ah | kWh | kg | € |
| 5 | 20 | 1.00 | 6.18 | 1.045 | 1.11 | 111.9 | 1.3 | 18.2 | 2178 |
| 6 | 20 | 1.20 | 7.42 | 1.045 | 1.34 | 134.3 | 1.6 | 21.9 | 2614 |
| 7 | 20 | 1.40 | 8.66 | 1.045 | 1.56 | 156.6 | 1.9 | 25.5 | 3049 |
| 8 | 20 | 1.60 | 9.90 | 1.045 | 1.78 | 179.0 | 2.1 | 29.2 | 3485 |
| 9 | 20 | 1.80 | 11.13 | 1.045 | 2.00 | 201.4 | 2.4 | 32.8 | 3920 |
| 10 | 20 | 2.00 | 12.37 | 1.045 | 2.23 | 223.8 | 2.7 | 36.5 | 4356 |
| 11 | 20 | 2.20 | 13.61 | 1.045 | 2.45 | 246.2 | 3.0 | 40.1 | 4792 |
| 12 | 20 | 2.40 | 14.84 | 1.045 | 2.67 | 268.5 | 3.2 | 43.8 | 5227 |
| 13 | 20 | 2.60 | 16.08 | 1.045 | 2.89 | 290.9 | 3.5 | 47.4 | 5663 |
| 14 | 20 | 2.80 | 17.32 | 1.045 | 3.12 | 313.3 | 3.8 | 51.1 | 6098 |
| 15 | 20 | 3.00 | 18.55 | 1.045 | 3.34 | 335.7 | 4.0 | 54.7 | 6534 |
| 16 | 20 | 3.20 | 19.79 | 1.045 | 3.56 | 358.0 | 4.3 | 58.4 | 6969 |
| 17 | 20 | 3.40 | 21.03 | 1.045 | 3.78 | 380.4 | 4.6 | 62.0 | 7405 |
| 18 | 20 | 3.60 | 22.26 | 1.045 | 4.01 | 402.8 | 4.8 | 65.7 | 7841 |

The daily walk distance of 30 km or more is currently economically suitable for using the PV charging system to fully cover consumption only by EV owners with consumption below 12 kWh/day (Table 8). No EV currently fulfills this condition.

**Table 8.** EV consumption limit value for optimal use of the PV system at a daily distance of up to 30 km/day.

| $EC_{EV}$ | $D_d$ | $EC_d$ | $A_{PV}$ | $H_{Md}$ | $P_{PV}$ | $Q_{bat}$ | | $M_{Bat}$ | $C$ |
|---|---|---|---|---|---|---|---|---|---|
| (kWh/100 km) | (km/day) | (kWh/day) | (m²) | (kWh/day) | kWp | Ah | kWh | kg | € |
| 5 | 30 | 1.50 | 9.28 | 1.045 | 1.67 | 167.8 | 2.0 | 27.4 | 3267 |
| 6 | 30 | 1.80 | 11.13 | 1.045 | 2.00 | 201.4 | 2.4 | 32.8 | 3920 |
| 7 | 30 | 2.10 | 12.99 | 1.045 | 2.34 | 235.0 | 2.8 | 38.3 | 4574 |
| 8 | 30 | 2.40 | 14.84 | 1.045 | 2.67 | 268.5 | 3.2 | 43.8 | 5227 |
| 9 | 30 | 2.70 | 16.70 | 1.045 | 3.01 | 302.1 | 3.6 | 49.3 | 5880 |
| 10 | 30 | 3.00 | 18.55 | 1.045 | 3.34 | 335.7 | 4.0 | 54.7 | 6534 |
| 11 | 30 | 3.30 | 20.41 | 1.045 | 3.67 | 369.2 | 4.4 | 60.2 | 7187 |
| 12 | 30 | 3.60 | 22.26 | 1.045 | 4.01 | 402.8 | 4.8 | 65.7 | 7841 |

In our search for economic optimism of the sustainability of the system, we analyzed the operating costs related to the operation of individual systems during monthly production in Slovakia. In the case of an oversized system (January to December), higher electricity generation and storage costs had to be considered, with accumulation costs in the order of magnitude. The results include the outputs of the analysis in the graphs Figures 5 and 6 for border accumulators marked in Table 5.

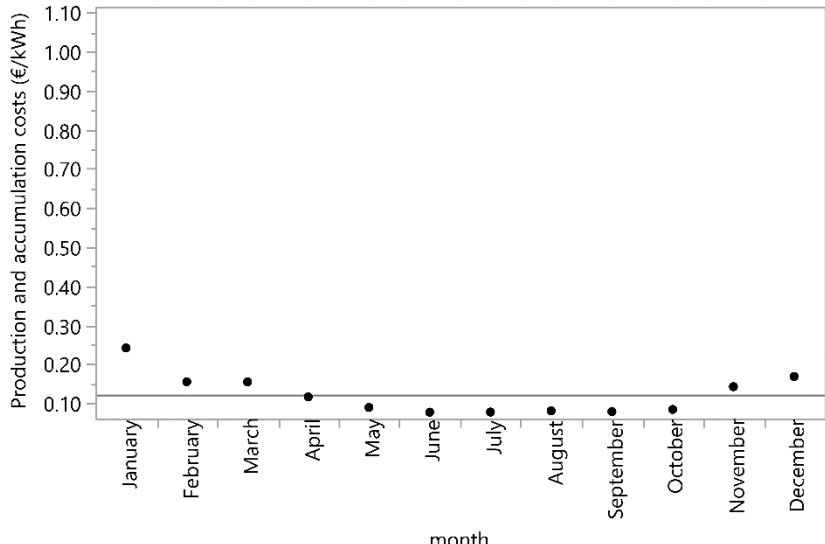

**Figure 5.** Unit costs for electricity generation and accumulation for LTO battery.

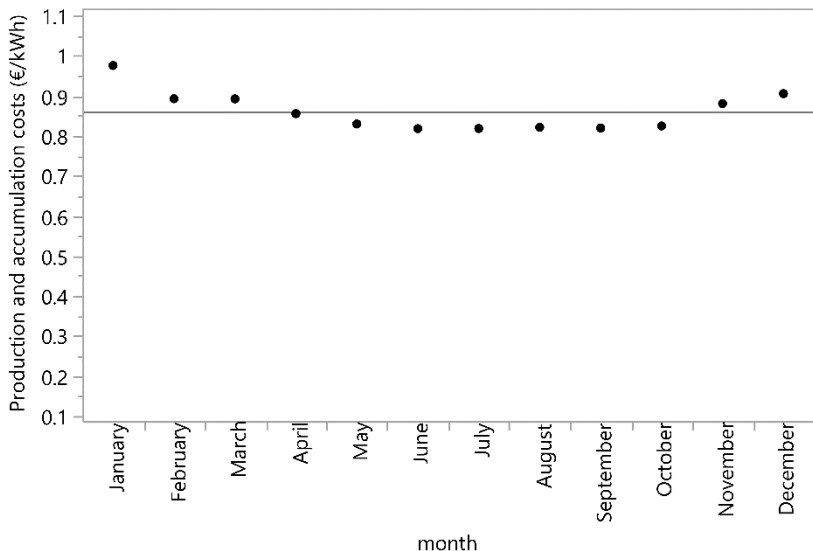

**Figure 6.** Unit costs of electricity generation and accumulation for Pb battery.

Unit operating costs have been determined variants by recalculating for a particular battery type all variants of the PV charging system with accumulation depending on EV consumption, daily distance traveled and given month.

$$CO = AP_{BAT} + \frac{C_{PV} \cdot P_{PV} \cdot 1000}{EC_d \cdot 365 \cdot 24} \quad (\text{€}/kWh) \tag{7}$$

Unit operating costs are composed of unit costs of accumulation $AP_{BAT}$ (Table 5), which depend on battery type and unit cost of the PV system ($C_{PV}$) in Slovakia. The current average price of PVs of the turnkey system without subsidies on the Slovak market is 2.4 €/kWp. The price is determined by its own market research between May and June 2020 [19–28].

It is evident that the number of battery discharge cycles has a significant impact on the unit price of electricity produced by the PV charging system with accumulation. For LTO battery (Figure 5) the unit cost of production and accumulation over the entire life of the system is more than 5 times lower than for the Pb battery (Figure 6).

We also analyzed the overall economic difficulty of the PV system, which is stated on Figures 7 and 8. The high cost of production and accumulation in case of system overloading is related to higher PV system performance and higher battery capacity. It is very important to choose a suitable battery size from the point of view of the economic demands of the whole system during its lifetime. According to [29–32], batteries can already be economically efficient without subsidies if the price of electricity does not rise more than inflation. The costs of operating the EV charging system in case of sub-dimension are related to the investment and maintenance of the PV charging system and the purchase of electricity from the grid. Excess or lack of electricity is expressed in Figures 7 and 8 blue, the economical difficulty of operating the system is red.

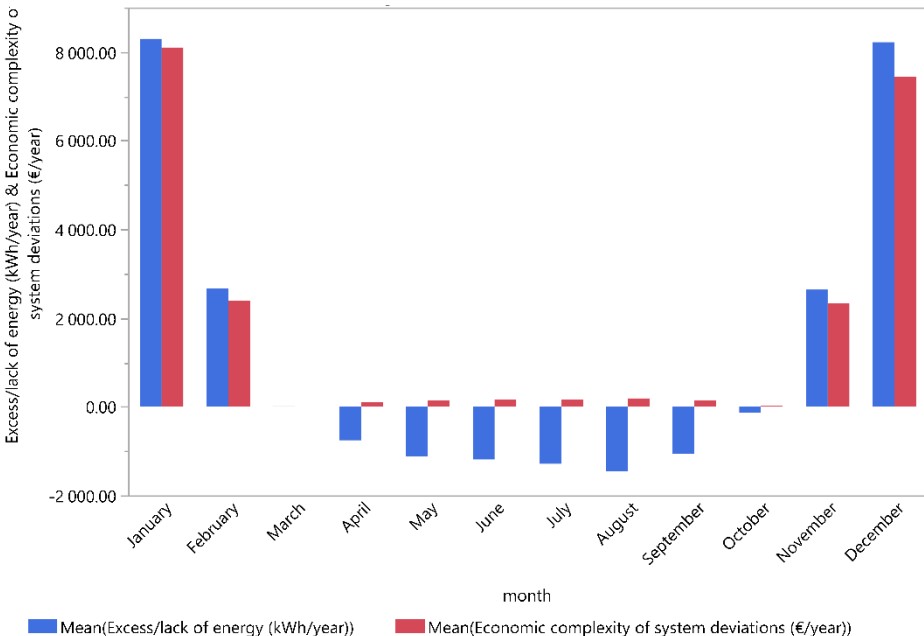

**Figure 7.** Impact of $f_{DM}$ on the energy and economic parameter of the system (LTO).

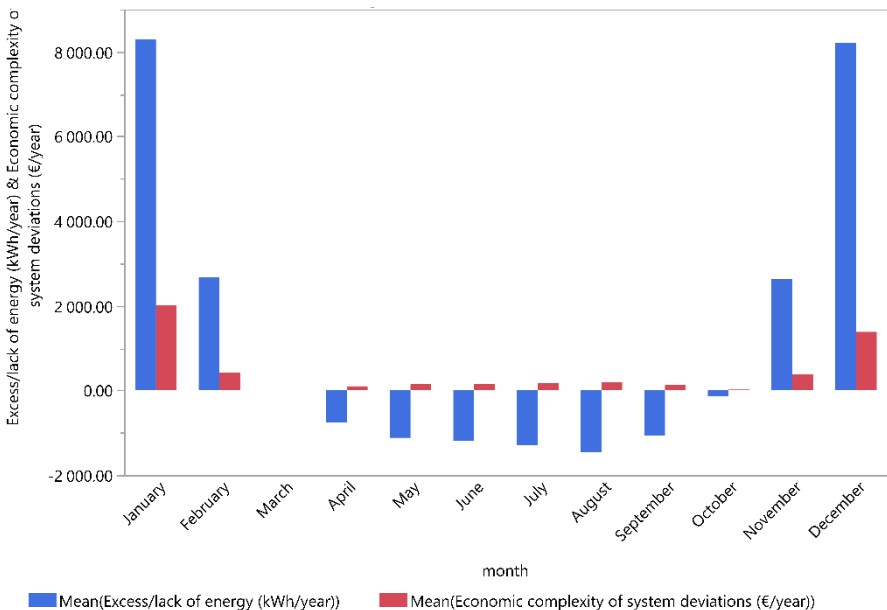

**Figure 8.** Impact of $f_{DM}$ on the energy and economic parameter of the system (Pb).

Compared to unit costs, the total cost of the PV charging system with accumulation over its lifetime for LTO batteries is 1/3 of the total cost of the PV system with Pb batteries. The length of the battery life and the need to replace it during the lifetime of the system was also considered in the analysis. The life of the inverter was also considered.

The economic difficulty marked in red for the PV system is significantly higher for full coverage in the winter months than in April-September. This means that, in these months, the economy of the PV charging system with accumulation is approaching the optimum or $f_{DM}$ is approaching 1.

In case of interest in supporting the charging of EVs with a PV system with an accumulation at current prices of technology and electricity in Slovakia, it is preferable to subdivision it and to cover insufficient production by purchasing electricity from the grid.

## 4. Conclusions

Solar energy is nowadays preferable as the additional source for EV charging, especially for two reasons: The PV panels are easier to install and effective than other RES, for example, wind energy. The energy production of PV panels is generated mainly during the highly priced grid tariff hours of the electrical grid. Thus, the EV charging stations can bring a new point of view on electricity prices from solar energy during peak hours.

Small photovoltaic systems (not power plants) are a common technology in the world, which has so far been used mainly as an additional source of electricity for households and companies. At present, however, they are perceived as an advantageous source of energy for transport. Electric vehicles represent the future of low-carbon transport, but only if the electricity they charge is made from clean sources. In the case of using a PV system in a family house, it is necessary to consider an economically advantageous mix of energy sources and harmonization of consumption over time. If this is not the case, a suitable energy accumulator must be used. Research [20] has found that if electricity and heat storage technologies, heat pumps, and battery-powered electric vehicles complement each other optimally, they will achieve the highest possible share of self-consumption for residential photovoltaic systems that can reach grid parity in this decade in most regions of the world.

Our research has shown that, by a suitable choice of battery, it is possible to achieve an economic optimum for PV-EV charging systems. However, it is necessary to consider as many input factors as possible, such as daily distance traveled and consumption from the EV point of view. From the battery selection point of view, it is necessary to focus primarily on the number of discharge cycles. When dimensioning the PV system, it is necessary to consider the current and future price of electricity from the grid and to find the so-called monthly deviation factor. This indicates the degree of oversizing or undersizing of the PV system in relation to the stated factors.

The computational model created within the presented research allows for analyzing a specific system consisting of EV, PV, and battery. The output is the stated $f_{DM}$ factor. The closer the $f_{DM}$ is to 1, the more optimal the designed system is in terms of the economics of its operation throughout its lifetime. The monthly deviation factor depends on the input variables valid for a specific country. In the case of Slovakia, it is suitable that the PV charging system with accumulation should be dimensioned so that $f_{DM} < 1$ is as close as possible to the value 1.

However, the economics of the combination of PV-EV and batteries alone cannot be just a matter of mathematical calculation or precision of the chosen technologies. As stated [20], the proper functioning of the PV-EV battery connection depends not only on technological progress or legislation but also on the extent to which the user accepts research results towards technology.

This means that it is necessary to educate users not only in the area of prices of individual technologies but especially in the area of using technology and consumer mixes. These will form the basis of energy concepts of all countries in the world in the near future [33,34]. When dimensioning the PV-EV battery system, it is therefore necessary to consider all variables not only in terms of investment but in terms of the lifetime of the entire system.

The proposed model for calculating the economic optimum of a PV charging system with energy storage is a suitable tool for predicting and verifying the design of a PV system for charging an electric car. Its advantage is that within the input data it is possible to substitute the values of PV technologies, solar radiation, batteries and economic parameters for any country. Currently, it is necessary to analyze the results of the created model in statistical software (we used JMP soft a.s., Bratislava, Slovak Republic). The aim of further research will be the integration of mathematical and statistical analyses into one generally applicable model.

At current energy and technology prices in Slovakia, a PV charging system with accumulation is economically acceptable only for EV owners who travel a distance of 20 km or less per day. This only applies if the PV system is used to charge the electric vehicle.

**Author Contributions:** Conceptualization, P.T., M.T., P.S., M.S.M., E.M.; methodology, P.T.; software, M.T.; validation, M.S.M., E.M.; formal analysis, M.T., E.M.; investigation, P.T., P.S., E.M.; resources, M.T., P.S., E.M.; data curation, M.S.M.; writing—original draft preparation, P.S.; writing—review and editing, P.T., P.S.; visualization, M.T.; supervision, P.T.; project administration, P.T. All authors have read and agreed to the published version of the manuscript.

**Funding:** This research was funded by Scientific Grant Agency of the Ministry of Education, Science, Research and Sport of the Slovak Republic (VEGA), grant number 1/0509/18.

**Conflicts of Interest:** The authors declare no conflict of interest.

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
