# Peer review of "Parameter Optimization Model Photovoltaic Battery System for Charging Electric Cars"

_energies, doi:10.3390/en13174497_

Round 1

Reviewer 1 Report

This article seems interesting and promising to me. However, there are several important drawbacks that should be corrected before publication. First and foremost is the recurrent and awkward appearance of the the phrase "Error. Reference source not found" from page 5 onwards which I find very misleading. Advancing on through the paper we find that radiation is very superficially studied as an average for Northern Europe at a latitude of 48 degrees North. As an expert in radiation, I suggest that more simulations be conducted in order to assess the value of solar radiation in much more detail for different locations. Otherwise the title should reflect that this procedure is only applicable in Slovakia and this it is limited. Moreover, the calculations presented in the article are very simple and static, based on quotients and formulas, I suggest that a more dynamic approach is introduced, trying to simulate several scenarios of probability depending on the different situations involved through variable modulators and fuzzy logic. In the conclusions a remark on the economic feasibility of the system is included but only for Slovakia, I consider this to be of limited scientific value as a more encompassing study should be required for ampler regions within Europe.

Author Response

I uploaded Response to the reviewer as an attachment in a Word file - 1. Review.edited.docx

Reviewer 2 Report

The paper presents a methodology for the design of a PV battery system for the charging of electric cars and apply it to the Slovakia case.

The methodology is interesting and the scientific soundness is sufficient for publication on Energies. However, there are several modification to be made to improve the quality of presentation that I’ll list below.

General corrections:

  • First of all, the document is full of “Reference source not found” related to equations, figures and tables that need to be fixed.
  • All symbols and acronyms (es. RE or t) must be explained the first time they are used.

Introduction:

  • The introduction must be improved with a better recognition of other works in literature on the same field;
  • Some terms like “energy batteries” and also entire statements ( “thermal energy does not require the huge outputs in short periods of time”) are quite obscure to me. Please explain better what you mean and revise the English form of the whole section using appropriate technical terms. E.g. line 74, pollutant are “emitted” not “transported” by internal combustion engines.
  • Actually, the authors should distinguish among tank to wheel greenhouse gases and tail pipe emissions that are specific of vehicles with internal combustion engines and other forms of emission (e.g. well to tank emissions of CO2, pollution associated to tire and brakes) which are also present in electric vehicles.
  • At the end of the introduction, the authors should better explain the novelty of their approach with respect to the state of the art in this field of research.

Materials and methods

  • The authors focus on the design of the charging station. What about the actual management of the different energy inputs and outputs?
  • The electric vehicle taken as reference for the minimum consumption (tesla Model 3) is not appropriate. Please enlarge the range of vehicles considered to this scope. See for example “Evaluation of emissions of CO2 and air pollutants from electric vehicles in Italian cities”, Applied Energy, Volume 157, Pages 675–687, (2015) DOI: 0.1016/j.apenergy.2014.12.089, ISSN: 03062619
  • The data of driving distance distribution depends strongly on the region of interest. Please specify the country to which the data of figure 1 refer to.
  • In the list of the battery parameters important for the EV owner, the authors cite the energy density. This leave me confused about which battery the authors are referring to. I do not understand, in particular, if the battery used as buffer in the proposed system is the that of the vehicles (like in ““An Integrated Tool to Monitor Renewable Energy Flows and Optimize the Recharge of a Fleet of Plug-in Electric Vehicles in the Campus of the University of Salento”, Proceedings of IFAC World Congress 2014, Vol. 19, Part 1, , pp 7861-7866, DOI: 10.3182/20140824-6-ZA-1003.01184, ISBN: 978-3-902823-62-5, ISSN: 1474-6670”) or a stationary battery included in the PV charging station. In the second case, the energy density of the battery is irrelevant.

Conclusions

  • Lines 396-411 are to be removed. In fact, conclusions should only summarize the methods adopted in the paper and the main results without figures, tables or references to literature.

Author Response

I uploaded Response to the reviewer as an attachment in a Word file - 2. Review.edited.docx

Reviewer 3 Report

The main remarks are:

  1. End of introduction: More information has to be given regarding your work. What are the main novelties of your paper? – (for example, Monthly production factor index, etc)
  2. Check the numbering of Sections and subsections, for example “3.1 Entry data and design of the computational model” has to change to 2.1, etc.
  3. In the considerations of PV efficiency, did you take into account the effect of temperature?
  4. Figures must be referenced in text. Moreover, in the majority of Figures, more explanation is needed.
  5. Similar remark for tables.
  6. Line 186: Please add reference. Check also the whole text for similar problems.
  7. Line 189 - battery parameters: Lifetime of battery (number of daily cycles) is also important for the owner, regarding economic analysis. Moreover, in your calculations, did you take into account the efficiency of the battery?
  8. Line 224: From which data the number 3120 comes from?
  9. Tables 3-4: You provide ECEV numbers that are outside the range of line 140.
  10. Figure 2: Please check if you can increase figure resolution. Moreover, more information has to be provided.
  11. Results and Conclusion sections have to be numbered.
  12. Line 166: nMD or HMD?
  13. The abbreviation PVS is not defined in text. Please check for similar examples.
  14. Check also that you define all parameters that can be found in equations in your manuscript.
  15. Line 59: “are renewable energy sources.”

Author Response

I uploaded Response to the reviewer as an attachment in a Word file - 3. Review.edited.docx

Round 2

Reviewer 1 Report

the phrase: Reference source not found.Error! Reference source not found.Error! 399 Reference source not found. continues to appear but may be at different positions of the manuscript.

I acknowledge the effort produced by the authors. I still maintain some minor doubts about the real scientific contribution provided to the field.

Author Response

Thank you for your comments.

Reviewer 3 Report

All remarks have been answered. It has to be noticed that in the revised manuscript there are a lot of error messages regarding references, which have to be corrected.

Author Response

Thank you for your comments.